structural engineering/artificial intelligence

convolutional neural networks, deep learning, design load, load rating, bridges

**Author for correspondence:**
Arya Pamuncak
e-mail: a.pamuncak@warwick.ac.uk

# Deep learning for bridge load capacity estimation in post-disaster and -conflict zones

Arya Pamuncak[1], Weisi Guo[1,2], Ahmed Soliman Khaled[1] and Irwanda Laory[1]

[1]School of Engineering, University of Warwick, Coventry CV4 7AL, UK
[2]Alan Turing Institute, British Library, 96 Euston Road, London NW1 2DB, UK

AP, 0000-0001-9829-2838; WG, 0000-0003-3524-3953

Many post-disaster and post-conflict regions do not have sufficient data on their transportation infrastructure assets, hindering both mobility and reconstruction. In particular, as the number of ageing and deteriorating bridges increases, it is necessary to quantify their load characteristics in order to inform maintenance and asset databases. The load carrying capacity and the design load are considered as the main aspects of any civil structures. Human examination can be costly and slow when expertise is lacking in challenging scenarios. In this paper, we propose to employ deep learning as a method to estimate the load carrying capacity from crowdsourced images. A convolutional neural network architecture is trained on data from over 6000 bridges, which will benefit future research and applications. We observe significant variations in the dataset (e.g. class interval, image completion, image colour) and quantify their impact on the prediction accuracy, precision, recall and $F$1 score. Finally, practical optimization is performed by converting multiclass classification into binary classification to achieve a promising field use performance.

## 1. Introduction

### 1.1. Bridge load testing

While government agencies typically maintain a database of the status of bridges and infrastructure, post-disaster and post-conflict scenarios can often lead to the loss of this knowledge. As many of our bridges are ageing and deteriorating, it is essential to understand the structures' condition for safe usage, to improve maintenance efficiency and guide the investment of new assets. In assessing a bridge's safety, the load level that can be applied to the bridge during its lifetime should be regarded as one of the most important properties in the bridge maintenance since the presence of overweight vehicles can lead to a decrease

in the service life [1]. According to American Association of State Highway and Transportation Officials (AASHTO) 'Manual for condition evaluation and load and resistance factor rating (LRFR) of highway bridges', load rating is generally used as a parameter which defines the load carrying capacity of a bridge [2]. In obtaining load rating of a bridge, visual inspection and load testing are commonly performed [3]. While visual inspection is rapid, truck load tests are commonly performed to remove subjective bias. However, the test requires a pre-weighed truck, bridge instrumentation and bridge closure which can be slow and expensive. In addition to load rating, design load can be used as another parameter which represents a bridge's load carrying capacity. According to [2], design load describes the assumption of live load used when a bridge is designed. Therefore, the bridge condition will not affect the design load since this load level is only employed in the design phase of a bridge. In [4], the design load has been classified into 13 classes and these classes represent load levels that are used in the design step of a bridge in the USA. There are clear limitations related to the aforementioned activities such as availability of resources, subjectivity, cost and time. This is exasperated in post-disaster zones, where expertise is expensive and sparse, and the need to rapidly understand the capacity of bridges is critical for recovery. In this situation, end users (i.e. aid workers and military) have to prioritize which bridges to inspect with limited resources. Therefore, we are motivated to classify broad bridge capacity values as a precursor to closer condition inspection.

## 1.2. Deep learning

During the past decade, scalable machine learning techniques that can process large amounts of high-dimensional data have transformed many sectors, such as automated driving, medical imaging and natural language processing. In particular, deep learning methods have replaced conventional feature-based machine learning by automating the feature extraction process and reducing the requirement of human domain expertise [5,6]. Deep learning achieves this by employing many layers of processing stages hierarchically [6–8]. Currently, deep learning has gained popularity in the image processing field, particularly due to the rise of convolutional neural networks (CNN). CNN has architectures which are inspired by part of mammalian brain which is known as the primary visual cortex in processing visual input [8], and it can be seen as the first successful deep learning architecture [6,7]. Introduced in the early 1990s, CNN rose into prominence in 2012 when a CNN-based prediction model produced the best performance in ImageNet competition. Following this success, CNN has revolutionized the field of computer vision and become a state of the art for recognition and detection purposes [6]. In addition, it has been used in a wide range of applications such as image classification for a large number of classes [9–11], traffic sign recognition [12], medical object classification [13–15], face recognition [16,17] and damage detection in structures [18–21]. In addition to automatic feature extraction, CNN can produce excellent performance for complex image recognition tasks due to the capability of CNN in exploiting the local spatial correlation between pixels in the image [6]. Unlike other image recognition algorithm, CNN depends on the spatial separation instead of the spatial position; hence, combination of local features is more important than the location of features which might be varied on images. This promising achievement and the ubiquitous use of mobile cameras have motivated the research in the development of a deep learning-based method for load rating estimation using images.

## 1.3. Machine learning in civil engineering maintenance

In general, a number of advanced computing methods have been developed over the last decades for monitoring and maintenance of civil infrastructure, including conventional vibration-based methods [22,23] and blind signal separation methods [24,25] as well as Bayesian approaches [26,27]. Recent research efforts have been made in automatic defect detection by using CNN in image processing. In 2017, Cha *et al.* [18] employed CNN technique for crack detection on concrete surfaces. In this research, 40 000 images were used to train a neural network and 55 testing images were used to observe the network performance. Comparative study with traditional Canny and Sobel edge detection method was conducted to observe the performance of the proposed method, and it was shown that the proposed system produces better capability in sensing thin cracks. Furthermore, Protopapadakis *et al.* [19] proposed an automatic robotic inspector for tunnel condition monitoring in 2016. In his work, CNN was employed for visual inspection of the robot. By using CNN, high-level discriminative features for complex nonlinear pattern classification were produced. These features later were used to calculate real-time three-dimensional (3D) information to identify the crack position and orientation. In addition,

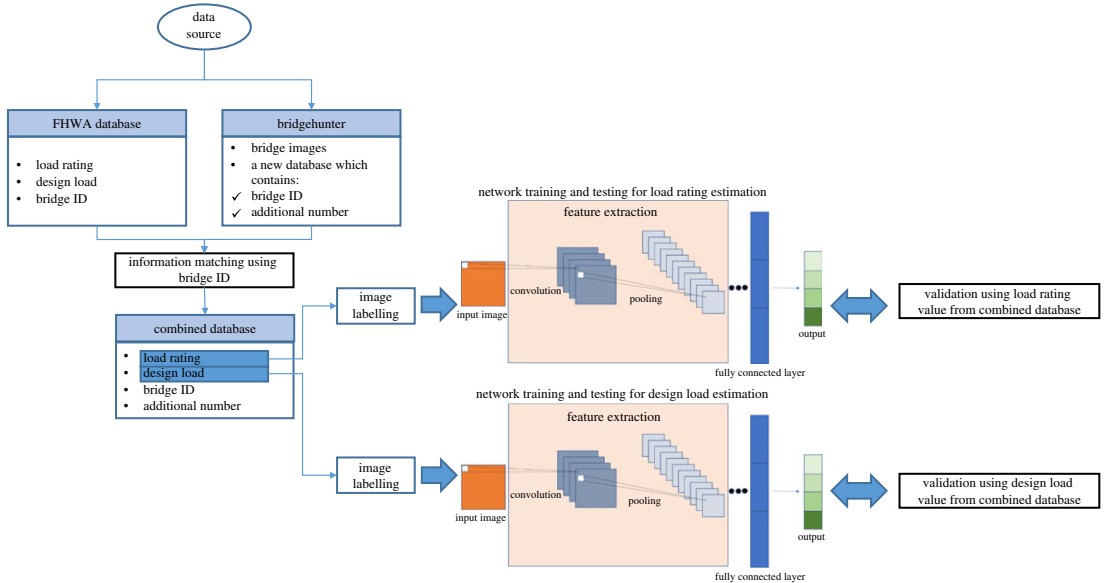

**Figure 1.** Schematic of data collection, data labelling, network testing and training and evaluation.

studies for crack detection on pavements using CNN have been conducted [20,21]. In these studies, pavement images were employed to train neural networks. However, unlike [20] which trained the neural networks from scratch, in [21], transfer learning using a pre-trained VGG-16 network was performed. Therefore, in the research, a pre-trained model was fine-tuned for the new prediction task. Both studies managed to detect the presence of a crack in the pavement images. Despite progress having been made in the implementation of deep learning for infrastructure condition monitoring, this implementation is limited to detection of defect. In addition, in the image processing field, quantifying bridge condition from its image still remains a challenging task [28]. Furthermore, the research mentioned previously employed images which were taken specifically for the research. In addition, others can also label images using human experts. In this study, images are downloaded from crowd and no expert is involved in the process. This can further provide a significant challenge in predicting a bridge's load carrying capacity.

## 1.4. Summary of novelty and contribution

This study is primarily aimed at addressing the global development challenges where bridge capacity data are missing and hindering reconstruction. Current methods rely on human expertise through either visual inspection or testing. However, this is expensive, slow and reliant on the availability of expertise. In this paper, by using deep learning as an automatic structure identification method, we have created a cheap, scalable, transparent and verifiable bridge load identification tool. In time, this method can be standardized and used on smartphones by non-experts. The proposed method will require no physical model of a bridge and thus can provide solution for countries that are lacking in civil engineering experts or bridge documentation. In addition, the trained CNN can benefit other researchers through transfer learning.

# 2. Methodology

This research proposes an alternative method for estimation of load carrying capacity of bridges. This method employs crowdsourced bridge images as an input and provides an estimation about either its load rating or design load, as a proxy for the load carrying capacity. The methodology performed in this research is shown in figure 1. The main activities that were performed in this project:

1. data collection of bridge images and load ratings of the bridges,
2. neural network training and testing, and
3. evaluation of performance and optimization.

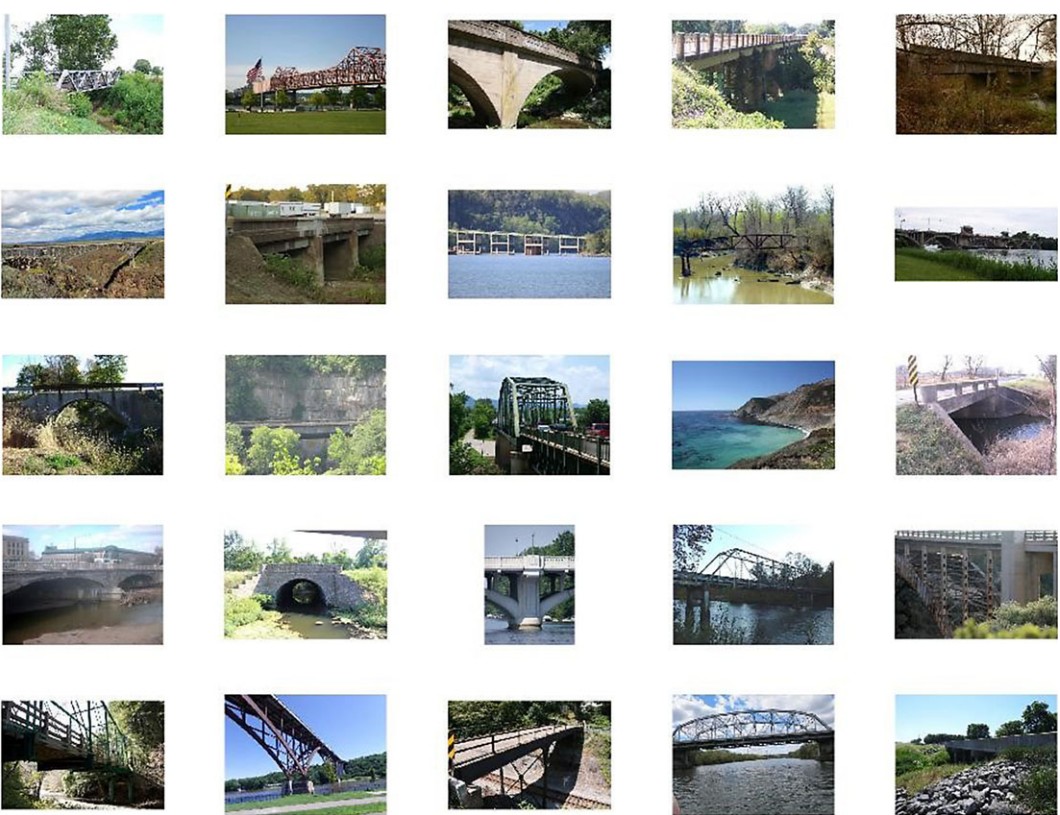

**Figure 2.** Images collected from web scraping.

## 2.1. Data collection and preparation

In this part of our study, the bridge database of load ratings and corresponding crowdsourced bridge images were collected. For the load ratings, the National Bridge Inventory (NBI) database published by American Federal Highway Administration (FHWA) [29] was employed. This database provides information about all bridges in the USA. Information such as inventory number, location, features, design load, construction, condition and load rating of bridges can be retrieved from this database. Some of this information was used in labelling bridge images. To obtain crowdsourced images for training the CNN, web scraping was conducted. In this part of the research, an interface program for collecting images from the web was built using Python. By using the program, 54 458 bridge images from 6753 bridges were collected from a website called www.bridgehunter.com [30]. In addition, inventory number and the state ID of these bridges were also collected since they were used as the matching information with the NBI database. To give an identifier to these images, an additional ID number was assigned to each bridge. This additional number was used as the folder name for the images. The datasets containing the bridge images can be obtained from [31]. Figure 2 shows samples that are collected for this research.

In obtaining either the load rating or design load information for the labelling step of the bridge images, both bridge inventory number and state ID were used as the matching information. State ID was used since bridges with similar inventory number can be found in other states. Hence, by using this method, information about load rating and design load for each image was obtained from the NBI database. However, it is difficult to apply this method to some bridges due to the difference in data format in certain states. In this case, the data in the NBI database have to be processed before they can be used as matching information with the data from bridgehunter database. Table 1 shows the number of samples according to the availability of the load carrying capacity information. Noted from table 1, there are some samples that have no label. This condition occurs due to the unavailability of either load rating or design load data for the corresponding bridges in the NBI database. Due to this problem, some samples can only be used for developing either load rating or design load prediction model.

**Table 1.** Summary of image attributes used for load rating and design load prediction.

| remark | no. images | images showing a full picture of a bridge | no. images only shows part of a bridge |
|---|---|---|---|
| all data | 54 458 | n.a. | n.a. |
| images for design load estimation | 18 821 | 7837 | 10 984 |
| images for load rating estimation | 43 180 | 13 510 | 29 670 |

## 2.2. Network training and testing

Instead of training a neural network from scratch, in this research, a transfer learning method was performed. In this way, a pre-trained network is fine-tuned in order to suit our application. The implementation of transfer learning using pre-trained networks such as AlexNet, VGG-16 and GoogLeNet has shown a great potential in solving a wide domain of image classification purposes [14]. It has been shown that, certain networks created using transfer learning technique can produce better performance than networks that are built from scratch [14,15]. In this research, AlexNet [9] was used as a pre-trained network for transfer learning. AlexNet is a neural network that has been trained to classify 1000 objects. The network combines both feature extraction and classification function in one body. The feature extractor automatically obtains important features for load capacity prediction, and the classifier part of the network performs prediction of load capacity using the extracted features. For transfer learning, the output layer of the neural network was altered according to the number of classes in the dataset we were working with. It was performed by modifying the number of connections at the final fully connected layer to match the number of classes in the dataset. To obtain training and testing data, the original dataset was randomly split into 80% and 20% for training and testing data, respectively. In addition, pre-processing of images was conducted to modify the size of these images, so they might be fed into the neural network. After this pre-processing step, the size of all images was altered to $227 \times 227 \times 3$ to match the input size of AlexNet.

## 2.3. Performance evaluation

To evaluate the neural networks performance, common parameters such as accuracy, precision, recall and $F1$ score were measured for each prediction model. The precision, recall and $F1$ score are given by (2.1)–(2.3), respectively [32]

$$\text{precision} = \frac{\text{TP}}{\text{TP} + \text{FP}}, \tag{2.1}$$

$$\text{recall} = \frac{\text{TP}}{\text{TP} + \text{FN}} \tag{2.2}$$

and

$$F1 \text{ score} = 2 \times \frac{\text{precision} \times \text{recall}}{\text{precision} + \text{recall}}. \tag{2.3}$$

where TP is true positive, FN is false negative and FP is false positive.

## 2.4. Dataset modification

In this research, dataset variation and its impact to the prediction performance were studied. Variations investigated in this research were the variation in the number of classes, bridge images completion and colour. The performance of the prediction model trained using these datasets was investigated.

### 2.4.1. Variation in prediction class

As it has been mentioned previously, the proposed system uses a bridge image to provide estimation about either its load rating or design load and this estimation is given in range of loading. This range of loading is defined by prediction classes, and classification task is performed by training prediction models. In this part of the research, CNN-based prediction models were trained by varying the number of classes for classification. This was performed to find the dataset configuration which produces the highest performance in predicting either load rating or design load of bridges. In addition, the effect of imbalanced database was also investigated. For this purpose, a number of datasets were generated. In

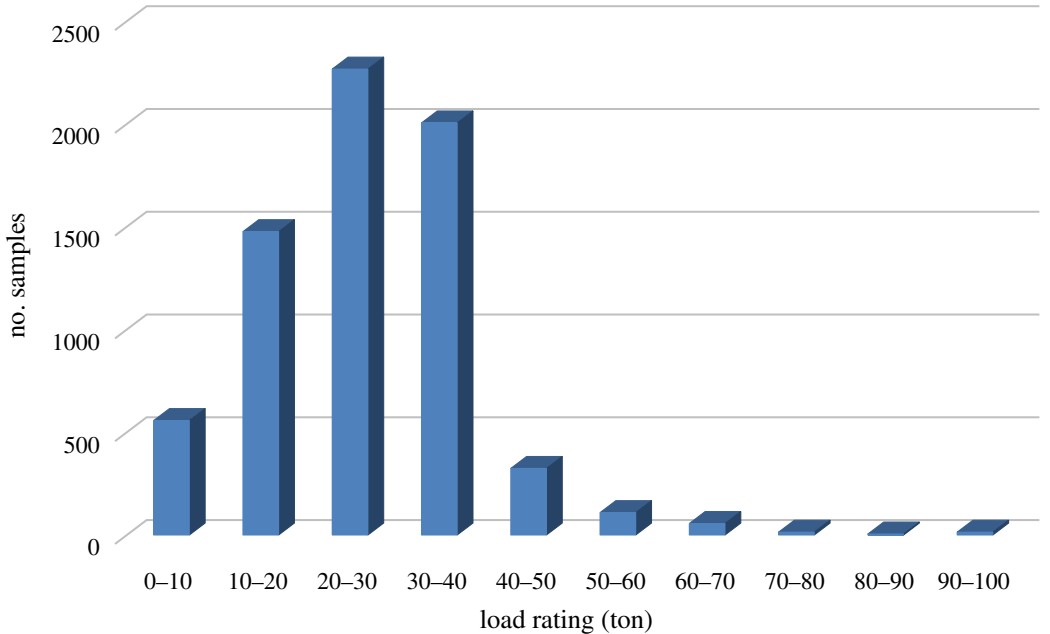

**data distribution according to load rating**

**Figure 3.** Data distribution according to the load rating.

our study, LRx represents datasets that are labelled with load rating information, while DLx represents datasets that are labelled using design load information.

Data distribution of samples according to their load rating can be seen in figure 3. In this research, estimation of load rating was performed by first discretizing the load rating value to obtain class labels. By using these labels, multiclass classification was performed for the estimation. To create dataset variation for load rating prediction, the class range was modified. In addition, the class interval modification took into account the number of samples obtained in each class. If one class was only formed by small number of images, this class was combined with the class adjacent to this class. As an example, in both dataset LR1 and LR2, due to the small number of samples with 0–5 ton of load rating, the 0–5 ton class was merged with the 5–10 ton class. From this modification, datasets were created and can be seen in table 2.

As is shown in figure 3, the data obtained for this research create an imbalanced dataset. Therefore, for each dataset in table 2, a balanced dataset is produced by having a down-sampling process of the majority class in the respective imbalanced dataset. These balanced and imbalanced datasets were used to train the prediction model, and the impact was observed.

Unlike the load rating data, the design load information has already been discretized in the NBI database. In developing the design load prediction model, several datasets were used by varying the number of classes used for the prediction. These datasets are shown in table 3. Dataset A was obtained as the original dataset for design load prediction. The class number was slightly modified from the class number given in [4]. It was done to sort these classes in an ascending order. This dataset consists of 12 classes, as shown in table 3; this dataset has imbalanced dataset where a lot of samples are obtained for classes 2, 3 and 5 and only few samples obtained in other classes, especially for classes 7, 8 and 11 which only have less than 100 samples. Hence, this dataset was not used in the research and other datasets were generated to overcome this problem. In DL1, classes 7, 8, 11 and 12 were removed from the dataset due to the limited number of samples. Therefore, this dataset only classified eight classes. In addition, to deal with data imbalance, in DL2, down-sampling of majority classes was performed. In this dataset, 1000 samples from each majority classes (category 2, 3 and 5) were randomly picked. Due to this process, this dataset only had 5774 samples.

Unlike DL1 and DL2 that removed samples from classes 7, 8, 11 and 12, both DL3 and DL4 combined these samples into one class. Therefore, these datasets were used to classify nine classes. DL3 used all samples while down-sampling process was applied on DL3 to generate DL4 in order to create more balanced dataset. Finally, both datasets DL5 and DL6 were created by combining samples from classes 5, 6 and 9 into one class. It was performed due to the similar load level applied to these classes.

**Table 2.** Description of datasets for load rating prediction.

| dataset | total no. classes | class description (ton) |
|---|---|---|
| LR1 and LR2 | 7 | 1. 0–10 |
| | | 2. 10–15 |
| | | 3. 15–20 |
| | | 4. 20–25 |
| | | 5. 25–30 |
| | | 6. 30–35 |
| | | 7. >35 |
| LR3 and LR4 | 5 | 1. 0–10 |
| | | 2. 10–20 |
| | | 3. 20–30 |
| | | 4. 30–40 |
| | | 5. >40 |
| LR5 and LR6 | 3 | 1. 0–15 |
| | | 2. 15–30 |
| | | 3. >30 |
| LR7 and LR8 | 3 | 1. 0–20 |
| | | 2. 20–40 |
| | | 3. >40 |

Similar to previous datasets, DL5 had imbalance sample distribution, while DL6 was generated as the balance version of DL5.

### 2.4.2. Design load prediction and load rating prediction models comparison

In order to create a fair comparison between load rating prediction model and design load prediction model, proprietary datasets were generated. In these datasets, the load rating class intervals were modified to match those in the design load. These generated datasets can be seen in table 4. In both DL7 and LR9, all images which can be labelled for either load rating or design load prediction were used. However, it can be seen in the table that there is a discrepancy between the number of images available for load rating prediction and the number of images applicable for design load prediction. In addition, it can be seen that both DL7 and LR9 are imbalanced datasets. Therefore, both DL8 and LR10 which have equal number of samples in every class were created. In order to reduce a prediction bias towards a majority class in imbalanced dataset, both DL9 and LR11 were generated. As can be seen from table 4, these datasets only have 300 samples in each class due to the small number of samples in class 6 available for design load prediction.

### 2.4.3. Image completion variation

The images obtained from bridgehunter website [30] contain both images showing a complete view of bridges and images showing incomplete bridge (only shows bridge connection, bridge railing, bridge deck or bridge column). This variation in the samples could affect the performance of the prediction model, therefore in this part of the research this impact was studied. Figure 4 shows samples that insufficiently represent a bridge. In order to filter these images, a CNN-based prediction was trained to classify whether or not an image is showing view of a bridge completely. From table 1, it can be seen that the number of images which show complete bridges is lower than the number of samples which show incomplete bridges. This can be seen as one of the limitations of this research since these images were obtained from the web; thus, it is challenging to control their quality. Several datasets were created: dataset which only contained samples showing complete bridges, dataset which only contained images showing incomplete bridge and dataset which contained all bridge images. To

**Table 3.** Description of datasets for design load prediction.

| class | metric | remark | dataset | | | | | | | |
|---|---|---|---|---|---|---|---|---|---|---|
| | | | A | DL1 | DL2 | DL3 | DL4 | DL5 | DL6 | |
| 1 | H10 | 10 ton | 928 | 928 | 928 | 928 | 928 | 928 | 928 | |
| 2 | H15 | 15 ton (3000 front 12 000 rear) | 4674 | 4674 | 1000 | 4674 | 1000 | 4674 | 1000 | |
| 3 | H20 | 20 ton (4000 front 16 000 rear) | 1913 | 1913 | 1000 | 1913 | 1000 | 1913 | 1000 | |
| 4 | HS15 | 27 ton (3000 front 12 000 mid and rear) | 460 | 460 | 460 | 460 | 460 | 460 | 460 | |
| 5 | HS20 | 36 ton (4000 front 16 000 mid and rear) | 3991 | 3991 | 1000 | 3991 | 1000 | 5067 | 1000 | |
| 6 | HS20+mod | equal to HS20 with the inclusion of military loading | 491 | 491 | 491 | 491 | 491 | 0 | 0 | |
| 7 | pedestrian | | 3 | 0 | 0 | 0 | 0 | 0 | 0 | |
| 8 | railroad | | 56 | 0 | 0 | 0 | 0 | 0 | 0 | |
| 9 | HL93 | equal to HS20 with an addition of road calculation | 585 | 585 | 585 | 585 | 585 | 0 | 0 | |
| 10 | HS25 | 45 ton or greater | 310 | 310 | 310 | 310 | 310 | 332 | 332 | |
| 11 | >HL93 | greater than HL93 | 22 | 0 | 0 | 0 | 0 | 0 | 0 | |
| 12 | other | for other bridge which employs other standard than AASHTO | 107 | 0 | 0 | 188 | 188 | 0 | 0 | |
| total | | | 13 540 | 13 352 | 5774 | 13 540 | 5962 | 13 540 | 5962 | |

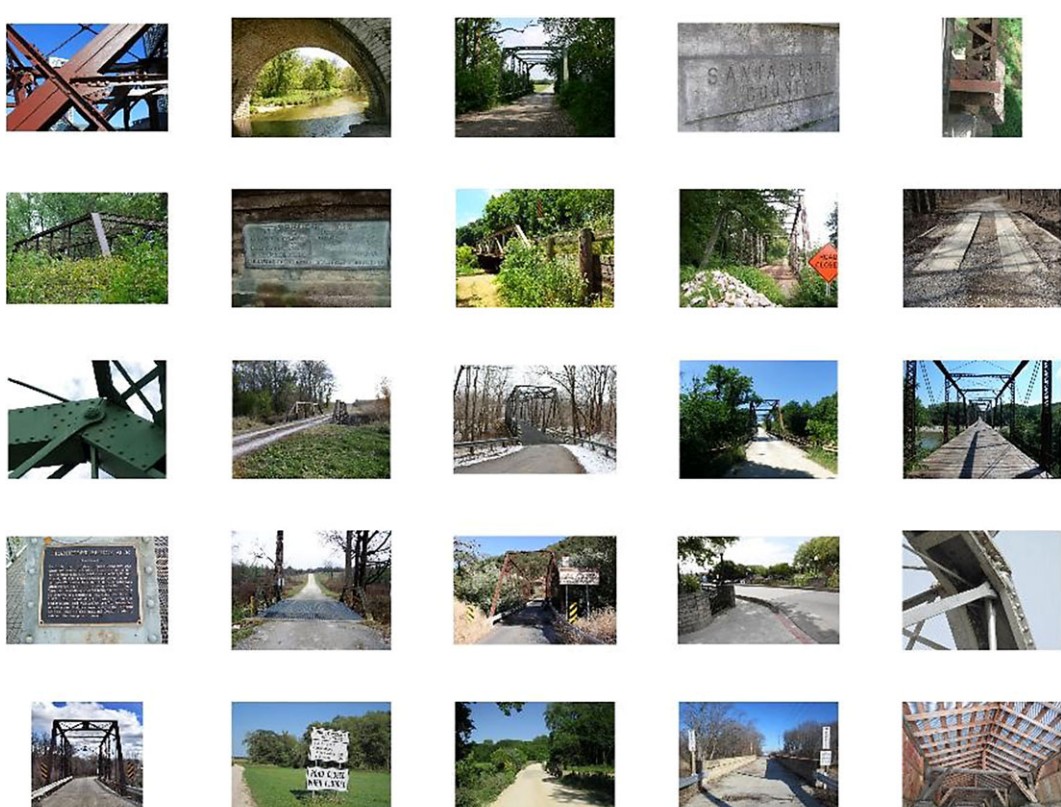

**Figure 4.** Images showing only part of a bridge.

**Table 4.** Description of datasets for comparison of load rating and design load prediction models.

| class | no. samples | | | |
| | DL7 | LR9 | both DL8 and LR10 | both DL9 and LR11 |
|---|---|---|---|---|
| 1 | 496 | 2098 | 496 | 300 |
| 2 | 2496 | 1153 | 1153 | 300 |
| 3 | 1201 | 1358 | 1201 | 300 |
| 4 | 287 | 2483 | 287 | 287 |
| 5 | 3171 | 2802 | 2802 | 300 |
| 6 | 186 | 920 | 186 | 186 |

make a fair comparison, these datasets were configured to have similar number of samples, hence down-sampling process was implemented. Table 5 shows the generated datasets.

In table 5, datasets A (DL10, DL11 and DL12) represent datasets which consist of both images showing complete and incomplete bridges, datasets B (DL13, DL14 and DL15) only include images showing complete bridges and datasets C (DL16, DL17 and DL18) only consist of images with incomplete view of bridge. The number described in each dataset shows data distribution on the datasets. Dataset DL10, DL13 and DL16 were datasets created directly from image filtering process based on the perspective of the images. However, as can be seen in table 5, there is a variation in the number of images inside these datasets. To create a fair comparison, the number of images in datasets A, B and C had to be equal; therefore, DL11, DL14 and DL17 were generated. As can be seen in table 5, these datasets have imbalance sample distribution and due to the equal number of samples in DL13 and DL14, basically DL14 is similar to DL13. To prevent prediction bias in imbalanced datasets, dataset DL12, DL15 and DL18 were produced. As shown in table 5, these datasets have almost equal number of samples in each class.

**Table 5.** Description of datasets for investigation of image completion to the performance of prediction model.

| class | dataset consisting all images (A) | | | dataset consisting only image with full view of bridges (B) | | | dataset consisting only images of incomplete bridge (C) | | |
|---|---|---|---|---|---|---|---|---|---|
| | DL10 | DL11 | DL12 | DL13 | DL14 | DL15 | DL16 | DL17 | DL18 |
| 1 | 1456 | 496 | 300 | 496 | 496 | 300 | 960 | 496 | 300 |
| 2 | 6389 | 2496 | 300 | 2496 | 2496 | 300 | 3893 | 2496 | 300 |
| 3 | 2635 | 1201 | 300 | 1201 | 1201 | 300 | 1434 | 1201 | 300 |
| 4 | 706 | 287 | 287 | 287 | 287 | 287 | 419 | 287 | 287 |
| 5 | 5406 | 2460 | 300 | 2460 | 2460 | 300 | 2946 | 2460 | 300 |
| 6 | 641 | 312 | 300 | 312 | 312 | 300 | 329 | 312 | 300 |
| 7 | 962 | 399 | 300 | 399 | 399 | 300 | 563 | 399 | 300 |
| 8 | 379 | 186 | 186 | 186 | 186 | 186 | 193 | 186 | 186 |

### 2.4.4. Image colour variation

This section was conducted in order to investigate the effect of image colour on the performance of prediction model. The greyscale images were obtained by simply converting colourful images into their greyscale versions. Colourful images are formed by a number of pixels and each colour pixel has combination of RGB colour space. In this research, luminance was implemented as the greyscale algorithm. Luminance is the standard greyscale algorithm and it has been used frequently by image processing software for computer vision tasks [33]. In addition, it has been implemented in other studies [34–36]. In Matlab, this algorithm is performed using 'rgb2gray' function. Greyscale conversion was performed by calculating the luminance which is defined by [37]

$$Y = 0.299R + 0.587G + 0.114B. \tag{2.4}$$

where $Y$ is the greyscale value, $R$, $G$ and $B$ are the red, green and blue intensity, respectively. From these greyscale images, a new dataset was created. This dataset was then employed to train a CNN-based prediction model. Finally, comparison between this prediction model and a prediction model created in the previous section was made to observe the effect of image colour to the prediction performance.

# 3. Results and discussion

## 3.1. Variation in number of classes

### 3.1.1. Load rating

Figure 5 shows the neural network performance for load rating prediction trained using eight datasets with modification in class interval. From the figure, it can be seen that an increase in the class interval yields a better performance. The maximum accuracy of 68.26% and precision of 60% are achieved on neural network trained using dataset LR7. However, the recall and $F1$ score produced by this network are slightly lower than those achieved by networks trained using dataset LR5, LR6 and LR8. In the figure, the minimum accuracy is obtained from neural network trained using dataset LR2. In addition, balancing the data using down-sampling method yields lower accuracy and precision. This decrease in accuracy from the effect of balanced dataset can be seen from figure 5. In the figure, it can be seen that for every imbalanced dataset (LR1, LR3, LR5 and LR7), the corresponding balanced dataset (LR2, LR4, LR6 and LR8) always produced lower accuracy. On the other hand, in terms of recall and $F1$ score, no similar trend occurs. However, the number of classes should be taken into account when evaluating the network performance. In the figure, accuracy higher than 60% is achieved on models trained using datasets that only have three classes. Therefore, accuracy higher than 60% is only achieved for three-classes prediction.

### 3.1.2. Design load

Figure 5 shows the performance of neural networks for design load prediction which are trained using various datasets. In the figure, it can be seen that for the imbalanced dataset scenario, neural network

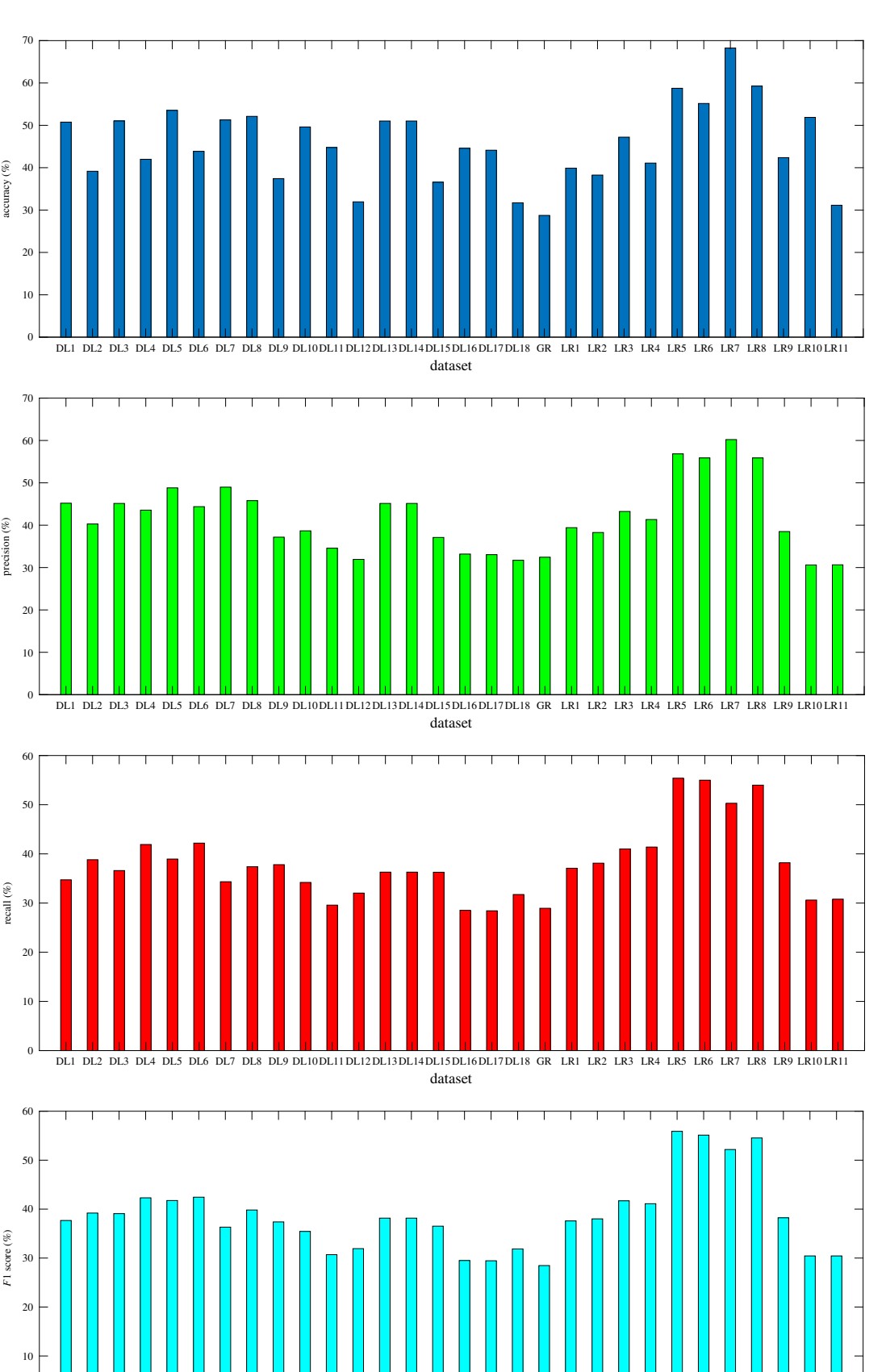

**Figure 5.** Performance of model trained for load rating or design load prediction.

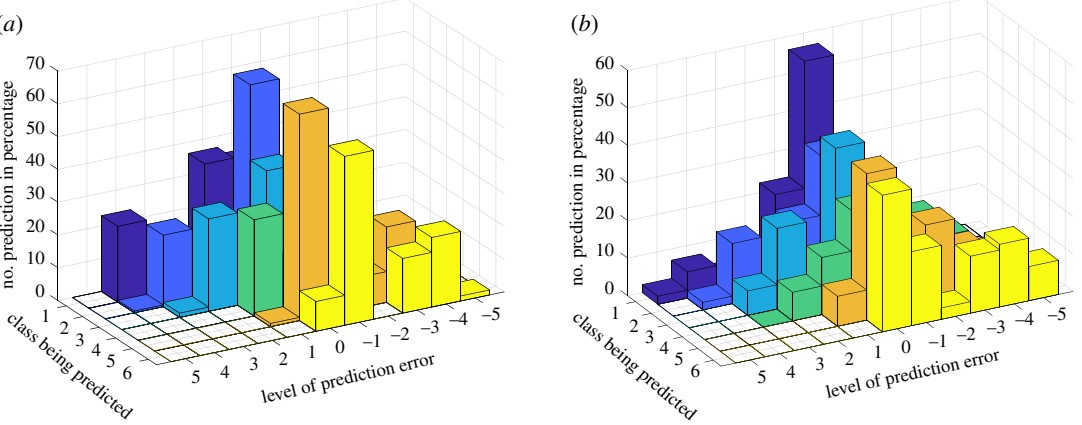

**Figure 6.** Distribution of error made by neural networks trained using imbalanced dataset (*a*) and balanced dataset (*b*).

trained using DL5 produces the highest performance. Compared with networks trained using DL1 and DL3, this neural network achieves higher accuracy, precision, recall and *F*1 score. On the other hand, on the balanced dataset scenario, neural network trained using DL6 performs best. It can be seen from the higher accuracy, precision, recall and *F*1 score that are produced by this neural network compared with those produced by neural networks trained using DL2 and DL4. In figure 5, the effect of balanced dataset can also be seen. From the figure, it can be seen that balancing the dataset has a negative relation with both accuracy and precision. This can be seen from every pair of balance and imbalance data (DL1–DL2, DL3–DL4 and DL5–DL6). However, it is also shown in figure 5 that balancing data can improve both the recall and *F*1 score.

In order to observe the effect of balanced dataset to the neural network's prediction, the prediction made by the networks was investigated. For this purpose, neural networks trained using DL5 and DL6 were investigated. Figure 6 shows the prediction of samples in every category which is made by neural networks trained using these datasets. From figure 6, it can be seen that although neural networks trained using DL6 produces lower accuracy than those trained using DL5, the bias towards majority classes is minimized. This condition can be seen in figure 6. In the figure, it can be seen that in imbalanced dataset, most prediction for samples from minority class is made towards majority class. On the other hand, for balanced dataset, almost in every class, maximum prediction on one class is made in the true class.

The higher accuracy obtained from neural networks trained using DL5 might be affected by the number of samples in the majority class. Although some misprediction was produced in the minority classes, the number of samples in majority class is much larger than the number of samples in minority classes as shown in table 3. Therefore, the true prediction made in the majority class contributes more than the misprediction; hence, higher accuracy can be achieved. In addition, unlike misprediction that is produced towards majority class in the imbalanced dataset scenario, from the figure, it can be seen that for balanced dataset, misprediction mostly occurs to the adjacent category. In this case, the prediction does not deviate too much from the true class. There are exceptions such as for samples in 15, 20 and 45 ton classes. This might occur due to the data distribution. Although down-sampling has been performed to balance the dataset, the number of samples in the down-sampled classes is still twice as much as the number of samples in some categories such as the 27 ton, 36 ton with military inclusion and 45 ton classes. Therefore, the bias prediction towards majority classes still occurs.

### 3.1.3. Comparison between models for design load prediction and models for load rating prediction

The performance of prediction models can be seen in figure 5. In the figure, it can be seen that for equal number of samples in the dataset, the models created for design load prediction perform better compared with models trained for load rating prediction. Comparisons are made between DL7 and LR9 (case 1), between DL8 and LR10 (case 2) as well as between DL9 and LR11 (case 3).

In case 1, all images available for either load rating prediction or design load prediction are used. In figure 5, it can be seen that DL7 produces higher accuracy and precision compared with LR9 and significant differences on these parameters are produced between these models. However, in case 1, the load rating prediction model produces slightly higher recall which leads to a slightly higher F1 score compared with the design load prediction model.

With equal number of images in each class (case 2), it can be seen that the design load prediction model produces higher performance than the load rating prediction model. This can be seen in figure 5 from the higher accuracy, precision, recall and $F$1 score obtained from the design load prediction model. Another interesting feature that can be observed in this figure is the higher accuracy achieved by the load rating prediction model compared with its performance on case 1. This event might occur due to the more imbalanced dataset used in case 2 compared with the dataset for case 1, which can be seen in table 5. In imbalanced dataset scenario, more predictions tend to be made on a majority class, which can lead to the increase in accuracy. However, the increase in accuracy does not represent a better performance, since in case 2 the load rating prediction model produce lower precision, recall and $F$1 score.

In balanced datasets scenario (case 3), it can be seen that the design load prediction model achieves higher accuracy, precision, recall and $F$1 score compared with the load rating prediction model. In this case, with similar number of data and minimum bias from imbalanced dataset, the design load prediction model still manages to outperform the load rating prediction model. Note that lower performance is produced in this scenario due to the small number of samples used in the datasets for case 3. Therefore, in all scenarios, the design load prediction models outperform the load rating prediction models.

## 3.2. The impact of image completion on the prediction performance

The performance of the prediction models trained using datasets with various image quality on three different scenarios can be seen from figure 5. For case 1 (DL10, DL13 and DL16), it can be seen that the neural networks trained using a dataset consisting of images which represent a bridge (DL13) perform the best of all. This can be seen from the highest accuracy, precision, recall and $F$1 score produced by this prediction model compared with the values from the other models. This performance is achieved using fewer samples in the dataset compared with other datasets. The number of data samples used in the training process might explain the slight difference between the accuracy achieved by the model trained using dataset DL10 and model trained using dataset DL13. In this case, dataset DL10 has more images compared with dataset DL13 as shown in table 5.

In case 2 (DL11, DL14 and DL17), where all datasets have an equal number of data samples in every class, it can be seen that prediction models trained using good quality images (DL14) produce the highest performance of all. Furthermore, since all datasets have an equal number of data samples, significant difference in the models' performance can be identified in this scenario. In addition, it is also shown that by using equal number of data samples, the model trained using images that do not completely show bridges achieves the lowest performance in terms of accuracy, precision, recall and $F$1 score.

In case 3 (DL12, DL15 and DL18), which is the balanced datasets scenario, it can be seen that the model trained using images with full view of bridges (DL15) also achieves the highest accuracy, precision, recall and $F$1 score among all prediction models. Similar to the previous section, decreases in performance were found in these prediction models. This might be due to the decrease in images number that are used to train these models because of the down-sampling process used to create balanced datasets.

In this comparison, the result shows that in all three cases, the models trained using good quality data (DL13, DL14 and DL15) produce the highest performance. From this comparison, it can be concluded that to obtain satisfactory performance, image quality plays an important role. This can be one factor that limits this research, since it is challenging to obtain images with the right angle from web scraping.

## 3.3. The impact of image colour on the prediction performance

Performance of prediction model trained using colourful images (DL6) and the performance of greyscale-trained prediction model (GR) are provided in figure 5. It can be seen that the implementation of greyscale images can worsen the performance of the load rating or design load estimation model. All parameters achieved by the model trained using greyscale image are lower than the parameters of prediction model trained using colourful images. This can be explained by the effect of colour in detecting a material where colourful image can give more information about the material that forms an object. Hence, it is more suitable to use colourful images for this kind of application.

## 3.4. Proposed optimization method for performance improvement

The result from prediction model shows unsatisfactory performance, which can be inferred from the low accuracy. In order to analyse this condition, a neural network for design load prediction trained using DL6 is taken as a case study. Figure 7 describes the training and validation process on a neural

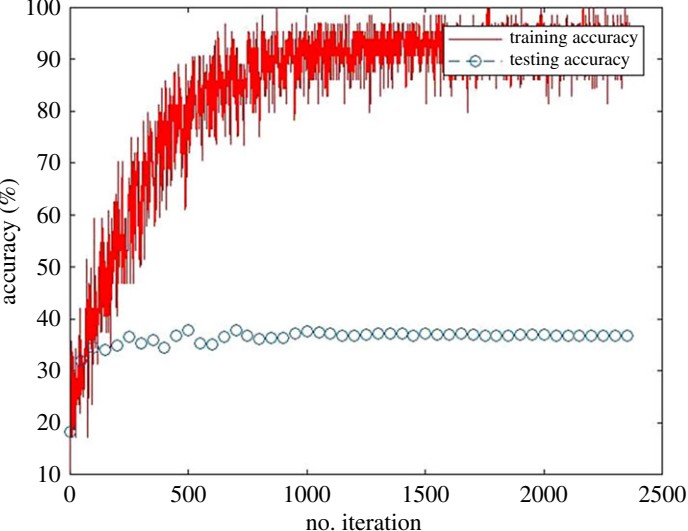

**Figure 7.** Training and validation accuracy of neural network for design load prediction.

network trained using this dataset. From the figure, it can be seen that although the training accuracy reaches 97% during the training process, overfitting occurs where no increase in the validation accuracy is produced after several iterations. Improvement of validation accuracy only occurs in the first 500 iterations before fluctuating validation accuracy is achieved. This leads to a significant difference between training and testing accuracy of the prediction model. This condition happens on all neural networks. Due to this situation, early stopping can be implemented to stop the training process whenever no increase in validation accuracy is achieved. However, no significant improvement is obtained only by using early stopping method. As can be seen in figure 7, early stopping can only slightly improve the accuracy to 41%, which is only 3% improvement from 38% produced without early stopping.

In order to observe the prediction error made by the model, an error probability distribution from the prediction is created. This error distribution is created by comparing the actual class and the prediction made using testing data. This error probability distribution is shown in figure 8. From the figure, it can be seen that the error follows normal distribution. In this case, most predictions are made into the true class and most error occurs when predicting a bridge into a class adjacent to the true class. Both load rating and design load prediction models produce similar trend in the error distribution.

From this error distribution, it is possible to increase the performance by merging two classes adjacent to each other. The simplest way is by converting the multiclass classification into binary classification. In this case, the model is not predicting the exact value of either load rating or design of a bridge in the image. However, the model is used to predict whether a bridge load rating or design load is higher or lower than a certain value. This conversion can be seen in figure 9.

Using binary classification conversion, the network is now used to predict:

— if a bridge has design load lower than 10 ton (level 1),
— if a bridge has design load lower than 15 ton (level 2),
— if a bridge has design load lower than 20 ton (level 3),
— if a bridge has design load lower than 27 ton (level 4),
— if a bridge has design load lower than 36 ton (level 5).

Note that from figure 9, before conversion, true prediction only occurs on the diagonal part of the confusion matrix that is identified in the green region. After the conversion, there are some regions that become either true positive (blue region) or true negative (light green region). By performing this conversion, improvement on prediction model can be obtained. The accuracy, precision, recall and F1 score produced by the system on each low level after conversion can be seen in figure 10. Following the conversion, it can be seen in figure 10 that an increase in the performance is produced. In addition, it is shown that the maximum accuracy, precision, recall and F1 score are produced when predicting in level 5. In this level, the accuracy, precision, recall and F1 score are 90.89, 95.21, 94.99 and 95.10 respectively.

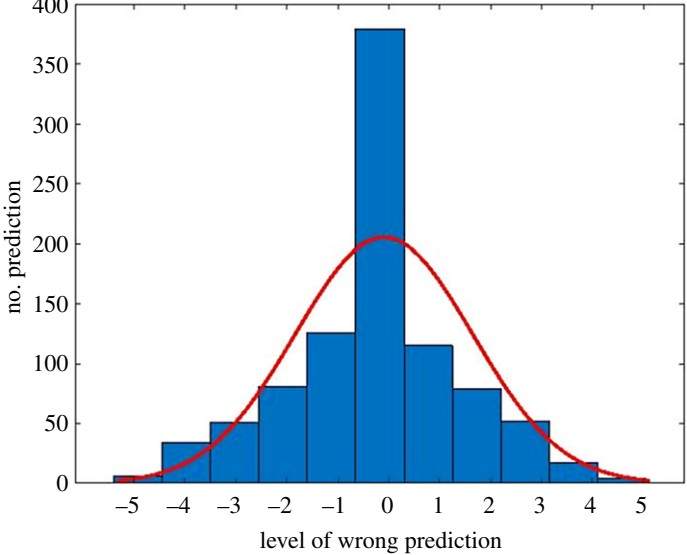

**Figure 8.** Error probability from design load prediction.

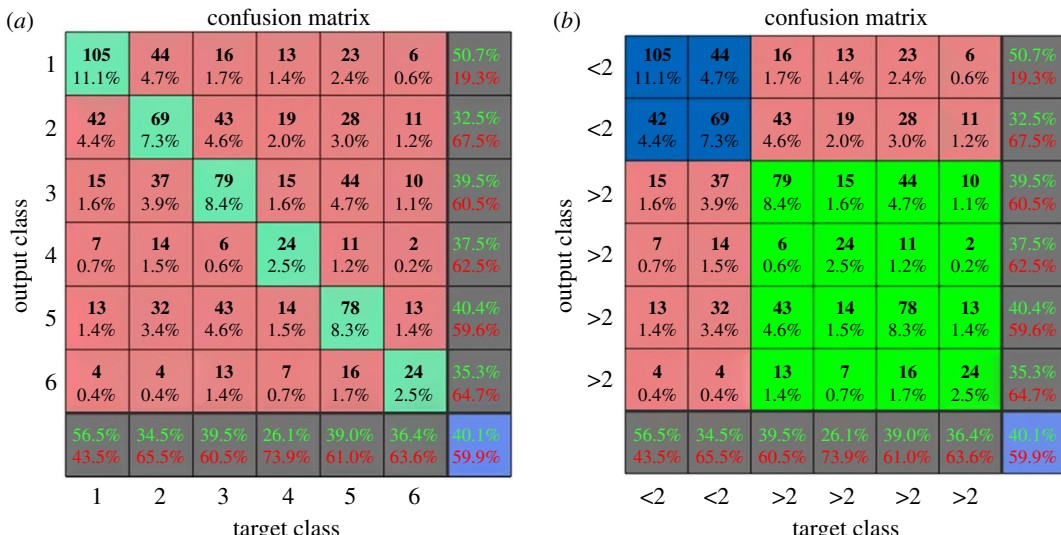

**Figure 9.** Conversion from multiclass classification into binary classification to detect if a bridge load rating is lower than 15 ton.

## 3.5. Analysis of the model's uncertainties

Certain factors might limit the applicability of our proposed method. Although this method might predict the bridge load rating using the features that can be extracted from images, it might be challenging to extract some important features from images. First of all, as has been discussed in §3.2, the perspective of a bridge in the image might influence the prediction result. This view might help in the extraction of features such as the bridge type, length and span. Hence, the view of the image should be taken into consideration when implementing this method.

In addition, there are other features which can be difficult to extract using images. This can be seen from the result in §3.1.3. In this section, it has been shown that predicting design load from bridge images is more achievable compared with predicting load rating from bridge images. Load rating of a bridge is affected by its condition, thus the bridge condition is one important feature for load rating estimation. However, it is a challenging task to quantify a bridge condition from its image. This situation might introduce error since the prediction models are unable to obtain this information from images. On the other hand, bridge condition provides no effect on the capacity which the bridge is designed for. This might lead to a better performance in design load prediction

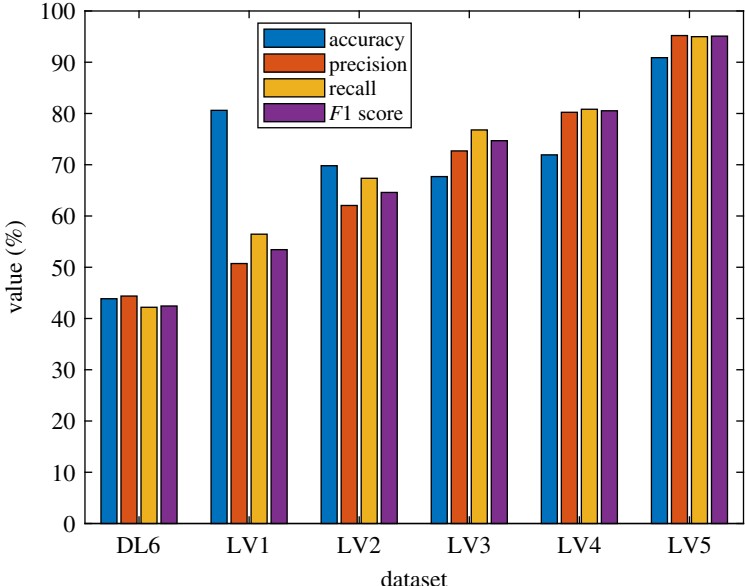

**Figure 10.** Performance of prediction model after conversion to binary classification.

model. Furthermore, at present, getting information of structural material from an image can be a challenge. This might be one of the sources of uncertainty in this study.

In addition to the presence of damage in the bridge, it is challenging to extract information inside the bridge using image processing. An example of this is determining whether a structure is a reinforced concrete structure or a pre-stressed concrete structure. The limitation of image processing in extracting this feature might introduce error for the proposed method. Therefore, to tackle this limitation, in our future work, we are planning to incorporate the features that are difficult to extract using image processing as additional inputs to the prediction models. Hence, along with the bridges' images, these features will be used as input data.

# 4. Conclusion

In this research, novel neural networks for estimating the load rating and design load of bridges have been trained based on crowdsourced image data. By using an equal number of samples, the neural networks trained for design load prediction produce higher performance than the neural network for load rating prediction. This can be seen from the higher accuracy, recall, precision and $F1$ score of the models trained for design load prediction compared with those of the load rating prediction models. The image quality affects the performance of prediction models. Therefore, in order to improve the performance, the quality of images used to train neural networks should be taken into account. It was found that the implementation of colourful images for this application is more suitable than using greyscale images. Furthermore, converting multiclass classification into binary classification can improve the prediction performance. Our future work will focus on refining the training process by artificially generating bridge data and images. In addition, to address the current limitation of image processing in detecting structure condition and gaining information about structural material, we are working in adding the information regarding bridge condition and material along with bridge images as the input of prediction model.

Data accessibility. Our data are available from the Dryad Digital Repository: https://doi.org/10.5061/dryad.6br51tn [31].
Authors' contributions. A.P. performed data collection, training and testing prediction model, and wrote the manuscript. I.L and W.G created the concept of the study, designed the study, coordinated the study and helped produce the manuscript, A.S.K. helped analyse the result and write the manuscript. All authors gave final approval for the publication of this paper.
Competing interests. W.G. is a Board Member of Royal Society Open Science.
Funding. This research was funded by Indonesian Endowment Fund for Education (grant no. PRJ-589 /LPDP.3/2017). The paper is co-funded by: Lloyd's Register Foundation's Data-Centric Engineering Program at the Alan Turing Institute (grant no. EP/N510129/1).
Acknowledgements. We would like to thank Prof. Jin Jiang and Prof. R. Kerry Rowe who have spent their efforts in reviewing our manuscript and provided insightful comments to help increase the quality of our manuscript.

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
