## [Reviewer comments · Royal Society Open Science]

Review History

RSOS-190227.R0 (Original submission)

Review form: Reviewer 1

Is the manuscript scientifically sound in its present form?

No

Are the interpretations and conclusions justified by the results?

Yes

Is the language acceptable?

Yes

Is it clear how to access all supporting data?

Yes

Do you have any ethical concerns with this paper?

No

Have you any concerns about statistical analyses in this paper?

No

Recommendation?

Major revision is needed (please make suggestions in comments)

Comments to the Author(s)

Deep learning is used to identify load ratings of bridges through a process of image classification. The premise appears to be that the capacity of a bridge can be obtained from an image that is taken from a distance away from the structure. This is flawed. While one can give a broad range for the load capacity based on the span length and structural make-up, it is difficult to actually assess condition from images, especially after an extreme event as mentioned as motivation. The authors need to give stronger arguments in support of their premise. There are also other issues as outlined below.

1. The authors discuss the colour of the images and the impact it can have on classification. More important, can be the perspective from which the image is taken. This is not discussed. Specifically there is little discussion on pre-processing the images and its impact on the classification.
2. The scientific link between the images and the classification is missing. How are the images input to the CNN? Is it a matrix of values? A particular concern is whether the CNN will provide a load rating for an image that does not contain a bridge.
3. There is no estimate of uncertainty in the predictions. The range of error in the predictions can actually render the prediction meaningless. There needs to be more calibrated testing to evaluate the uncertainty.
- 4, Figure 5 lacks clarity. Is it supposed to be a 3D plot?
5. Figure 7 Testing accuracy does not increase with iterations. 30% appears quite low.
6. Figure 8 Is the error plot for training or test data?
7. What are DL1, DL2 etc? These don't seem to be explained.

Review form: Reviewer 2**Is the manuscript scientifically sound in its present form?**

No

Are the interpretations and conclusions justified by the results?

No

Is the language acceptable?

Yes

Is it clear how to access all supporting data?

No

Do you have any ethical concerns with this paper?

No

Have you any concerns about statistical analyses in this paper?

Yes

Recommendation?

Major revision is needed (please make suggestions in comments)

Comments to the Author(s)

The method in this paper is the current practice of high-profile artificial intelligence. However, in bridge engineering, the application mainly focuses on bridge monitoring. It is not reliable to identify bridge load grade by deep learning. There are several questions for the author to answer: Firstly, most of the identified images cannot reflect the full view of the bridge, How to ensure the accuracy of identification.

Second, image recognition cannot judge the information inside the bridge, such as how to distinguish the reinforced concrete structure from the prestressed concrete structure. If not, how to ensure the accuracy only by image recognition.

Thirdly, in the operation stage, the bearing capacity of the bridge structure will decline to different degrees, which is not considered in the identification results.

Fourth, the author uses the data from NBI database for deep learning. If the results of deep learning are used to identify a bridge in other countries, what is the recognition result.

Fifth, what are the feature points for bridge identification.

Sixth, whether the technology has been successfully implemented, if so, please provide.

Decision letter (RSOS-190227.R0)

10-Jul-2019

Dear Mr Pamuncak,

The editors assigned to your paper ("Deep Learning for Bridge Load Capacity Estimation in Post-Disaster and -Conflict Zones") have now received comments from reviewers. We would like you to revise your paper in accordance with the referee and Associate Editor suggestions which can be found below (not including confidential reports to the Editor). Please note this decision does not guarantee eventual acceptance.

Please submit a copy of your revised paper before 02-Aug-2019. Please note that the revision deadline will expire at 00.00am on this date. If we do not hear from you within this time then it will be assumed that the paper has been withdrawn. In exceptional circumstances, extensions may be possible if agreed with the Editorial Office in advance. We do not allow multiple rounds of revision so we urge you to make every effort to fully address all of the comments at this stage. If deemed necessary by the Editors, your manuscript will be sent back to one or more of the original reviewers for assessment. If the original reviewers are not available, we may invite new reviewers.

When submitting your revised manuscript, you must respond to the comments made by the referees and upload a file "Response to Referees" in "Section 6 - File Upload". Please use this to document how you have responded to the comments, and the adjustments you have made. In

order to expedite the processing of the revised manuscript, please be as specific as possible in your response.

- Data accessibility

If you wish to submit your supporting data or code to Dryad (<http://datadryad.org/>), or modify your current submission to dryad, please use the following link:
<http://datadryad.org/submit?journalID=RSOS&manu=RSOS-190227>

- Competing interests

- Authors' contributions

- Acknowledgements

- Funding statement

on behalf of Professor Jin Jiang (Associate Editor) and R. Kerry Rowe (Subject Editor)
openscience@royalsociety.org

Comments to Author:

Reviewers' Comments to Author:

Reviewer: 1

Comments to the Author(s)

Deep learning is used to identify load ratings of bridges through a process of image classification. The premise appears to be that the capacity of a bridge can be obtained from an image that is taken from a distance away from the structure. This is flawed. While one can give a broad range for the load capacity based on the span length and structural make-up, it is difficult to actually assess condition from images, especially after an extreme event as mentioned as motivation. The authors need to give stronger arguments in support of their premise. There are also other issues as outlined below.

1. The authors discuss the colour of the images and the impact it can have on classification. More important, can be the perspective from which the image is taken. This is not discussed. Specifically there is little discussion on pre-processing the images and its impact on the classification.
2. The scientific link between the images and the classification is missing. How are the images input to the CNN? Is it a matrix of values? A particular concern is whether the CNN will provide a load rating for an image that does not contain a bridge.
3. There is no estimate of uncertainty in the predictions. The range of error in the predictions can actually render the prediction meaningless. There needs to be more calibrated testing to evaluate the uncertainty.
- 4, Figure 5 lacks clarity. Is it supposed to be a 3D plot?
5. Figure 7 Testing accuracy does not increase with iterations. 30% appears quite low.
6. Figure 8 Is the error plot for training or test data?
7. What are DL1, DL2 etc? These don't seem to be explained.

Reviewer: 2

Comments to the Author(s)

The method in this paper is the current practice of high-profile artificial intelligence. However, in bridge engineering, the application mainly focuses on bridge monitoring. It is not reliable to identify bridge load grade by deep learning. There are several questions for the author to answer:

Firstly, most of the identified images cannot reflect the full view of the bridge, How to ensure the accuracy of identification.

Second, image recognition cannot judge the information inside the bridge, such as how to distinguish the reinforced concrete structure from the prestressed concrete structure. If not, how to ensure the accuracy only by image recognition.

Thirdly, in the operation stage, the bearing capacity of the bridge structure will decline to different degrees, which is not considered in the identification results.

Fourth, the author uses the data from NBI database for deep learning. If the results of deep learning are used to identify a bridge in other countries, what is the recognition result.

Fifth, what are the feature points for bridge identification.

Sixth, whether the technology has been successfully implemented, if so, please provide.

Author's Response to Decision Letter for (RSOS-190227.R0)

See Appendix A.

RSOS-190227.R1 (Revision)

Review form: Reviewer 1

Is the manuscript scientifically sound in its present form?

Yes

Are the interpretations and conclusions justified by the results?

Yes

Is the language acceptable?

No

Do you have any ethical concerns with this paper?

No

Have you any concerns about statistical analyses in this paper?

No

Recommendation?

Accept with minor revision (please list in comments)

Comments to the Author(s)

The authors have addressed most reviewer comments. The main concern remaining is the language. There are a number of grammatical errors and these can be fixed with a careful proofread or with assistance from a native English speaker.

A secondary concern is regarding the main focus of the paper. The emphasis appears to be on an approach to approximately but rapidly load rate a bridge in the aftermath of a major event. While

this is mentioned to some degree, the authors could also make it clear that the proposed approach will not pick up localised or global damage but instead just attempt to quickly assess the load capacity based on a picture that provides information such as span length, width, etc. In fact, the information that needs to be in the picture can be discussed more in the study by relating to the performance of the approach and trying to come up with inferences on the information lacking in the pictures that were incorrectly interpreted.

Review form: Reviewer 2

Is the manuscript scientifically sound in its present form?

Yes

Are the interpretations and conclusions justified by the results?

Yes

Is the language acceptable?

Yes

Do you have any ethical concerns with this paper?

No

Have you any concerns about statistical analyses in this paper?

No

Recommendation?

Accept with minor revision (please list in comments)

Comments to the Author(s)

I am glad to see your detailed answers to the questions raised. It is suggested that the author make a brief analysis of the causes of errors in a separate section.

Decision letter (RSOS-190227.R1)

03-Sep-2019

Dear Mr Pamuncak:

On behalf of the Editors, I am pleased to inform you that your Manuscript RSOS-190227.R1 entitled "Deep Learning for Bridge Load Capacity Estimation in Post-Disaster and -Conflict Zones" has been accepted for publication in Royal Society Open Science subject to minor revision in accordance with the referee suggestions. Please find the referees' comments at the end of this email.

The reviewers and Subject Editor have recommended publication, but also suggest some minor revisions to your manuscript. Therefore, I invite you to respond to the comments and revise your manuscript.

- Ethics statement

- Data accessibility

<http://datadryad.org/submit?journalID=RSOS&manu=RSOS-190227.R1>

- Competing interests

- Authors' contributions

- Acknowledgements

- Funding statement

Please note that we cannot publish your manuscript without these end statements included. We have included a screenshot example of the end statements for reference. If you feel that a given

heading is not relevant to your paper, please nevertheless include the heading and explicitly state that it is not relevant to your work.

Because the schedule for publication is very tight, it is a condition of publication that you submit the revised version of your manuscript before 12-Sep-2019. Please note that the revision deadline will expire at 00.00am on this date. If you do not think you will be able to meet this date please let me know immediately.

on behalf of Professor Jin Jiang (Associate Editor) and R. Kerry Rowe (Subject Editor)
openscience@royalsociety.org

Reviewer comments to Author:

Reviewer: 1

Comments to the Author(s)

The authors have addressed most reviewer comments. The main concern remaining is the language. There are a number of grammatical errors and these can be fixed with a careful proofread or with assistance from a native English speaker.

A secondary concern is regarding the main focus of the paper. The emphasis appears to be on an approach to approximately but rapidly load rate a bridge in the aftermath of a major event. While this is mentioned to some degree, the authors could also make it clear that the proposed approach will not pick up localised or global damage but instead just attempt to quickly assess the load capacity based on a picture that provides information such as span length, width, etc. In fact, the information that needs to be in the picture can be discussed more in the study by relating to the performance of the approach and trying to come up with inferences on the information lacking in the pictures that were incorrectly interpreted.

Reviewer: 2

Comments to the Author(s)

I am glad to see your detailed answers to the questions raised. It is suggested that the author make a brief analysis of the causes of errors in a separate section.

Author's Response to Decision Letter for (RSOS-190227.R1)

See Appendix B.

Decision letter (RSOS-190227.R2)

02-Oct-2019

Dear Mr Pamuncak,

I am pleased to inform you that your manuscript entitled "Deep Learning for Bridge Load Capacity Estimation in Post-Disaster and -Conflict Zones" is now accepted for publication in Royal Society Open Science.

You can expect to receive a proof of your article in the near future. Please contact the editorial office (openscience_proofs@royalsociety.org and openscience@royalsociety.org) to let us know if

you are likely to be away from e-mail contact -- if you are going to be away, please nominate a co-author (if available) to manage the proofing process, and ensure they are copied into your email to the journal.

Best regards,
Lianne Parkhouse
Royal Society Open Science
openscience@royalsociety.org

on behalf of Professor Jin Jiang (Associate Editor) and R. Kerry Rowe (Subject Editor)
openscience@royalsociety.org

Appendix A

Dear Ms. Alice Power,

We would like to thank you for providing us this opportunity to revise our manuscript titled Deep Learning for Bridge Load Capacity Estimation in Post-Disaster and -Conflict Zones to *Royal Society Open Science*. The feedback and suggestions offered by the reviewers have been constructive, and we appreciate the time and effort that you and the reviewers have dedicated for providing us this valuable feedback. We have addressed the reviewer comments in our revised manuscript, and we have highlighted the change within the manuscript. Please find our responses in point-by-point manner below.

Reviewer: 1

Comments to the Author(s)

Deep learning is used to identify load ratings of bridges through a process of image classification. The premise appears to be that the capacity of a bridge can be obtained from an image that is taken from a distance away from the structure. This is flawed. While one can give a broad range for the load capacity based on the span length and structural make-up, it is difficult to actually assess condition from images, especially after an extreme event as mentioned as motivation. The authors need to give stronger arguments in support of their premise. There are also other issues as outlined below.

Response:

We strongly agree with you that it is challenging to assess bridge condition from image since the bridge condition might be have an important role in obtaining the load rating.

As such, we have revised the premise of our paper (introduction) to state more clearly: *“In this situation end users (i.e., aid workers and military) have to prioritise which bridges to inspect with limited resources. Therefore, we are motivated to classify broad bridge capacity values as a precursor to closer condition inspection.”* We also go on to say that our future work intends to combine the proposed generalised classifier with conditional monitoring classification which we are currently working on.

1. The authors discuss the colour of the images and the impact it can have on classification. More important, can be the perspective from which the image is taken. This is not discussed. Specifically there is little discussion on pre-processing the images and its impact on the classification.

Response:

We strongly agree with your recommendation that the perspective from which the image is taken might have an impact to the prediction model’s performance. Therefore, we have also observed this impact in our study. The result is reported in section 4.2 (page 7) and the methodology is written in section 3.4.3 (page 5). In this part of the study, some datasets have been generated. These datasets are grouped into three categories:

- (1) datasets consisting all images,
- (2) datasets consisting images showing full view of bridges, and
- (3) datasets consisting images showing incomplete view of bridges.

Example of images that shows incomplete view of bridges is in Figure 4. To study the impact of image’s perspective to the prediction’s performance, three cases were applied. In case 1, all images available for each category are used. Therefore, the number of images in each category is varied. In case 2, the number of samples for all categories are made to be equal in order to create a fair comparison. Finally,

in case 3, balance datasets for all three categories are generated in order to remove bias toward majority classes. In all three cases, it has been found that the prediction model that is trained using images that shows complete view of bridges produces the highest performance.

In our study, there are only two pre-processing steps performed on the image. The first pre-processing step is performed in the alteration of image size. This step is conducted in order to configure all images so they can be fed into the CNN. We have modified section 3.2 (page 3) in the revised manuscript to include this information. The second step is performed in converting colourful images into greyscale images. In this research, we implement luminance method as the grayscale algorithm. Luminance is the standard grayscale algorithm and it has been utilised frequently by some image processing software for computer vision tasks [1]. In MATLAB, this algorithm is performed using "rgb2gray" function. This information has been added in section 3.4.4 (page 5) of the revised manuscript.

2. The scientific link between the images and the classification is missing. How are the images input to the CNN? Is it a matrix of values? A particular concern is whether the CNN will provide a load rating for an image that does not contain a bridge.

Response:

The input of our prediction model are images that are formed by individual pixels. These individual pixels are defined by the combination of red, blue, and green component that form three-dimensional matrix. The matrix is the input of the CNN. Additional explanation has been given in section 3.2 (page 3). Regarding the input of the CNN. The current prediction model will give prediction for an image that does not contain bridge. However, it can be tackled by adding another label in the output layer of the CNN therefore all images that do not contain bridge will be predicted in that new class. We believe from an end user perspective, they will only input images with the targeted bridge.

3. There is no estimate of uncertainty in the predictions. The range of error in the predictions can actually render the prediction meaningless. There needs to be more calibrated testing to evaluate the uncertainty.

Response:

In order to observe the uncertainty in the predictions, we have generated and studied the error distribution that is created by our model. In our manuscript, this error distribution is described in Figure 8. The table below shows the corresponding error distribution. In the table, the level of wrong prediction describe how far the wrong prediction deviate from its true class. Negative level represents a condition where the prediction is lower than the true class and positive level occurs when prediction is higher than actual class. True prediction produce 0 level of wrong prediction. From the table, it is shown that despite the errors, most predictions are made into the right classes (379). In addition, most error occur when prediction is made to the class adjacent to the true class (126 and 115). In this case, the wrong predictions do not deviate far from their true classes.

Level of wrong Prediction	Prediction Made
-5	6
-4	34
-3	51
-2	81
-1	126
0	379
1	115
2	79
3	52
4	17
5	4

4. Figure 5 lacks clarity. Is it supposed to be a 3D plot?

Response:

Thank you very much for your feedback regarding to the clarity of Figure 5. Figure 5 is a 3D plot. It has been modified in the revised manuscript.

5. Figure 7 Testing accuracy does not increase with iterations. 30% appears quite low.

Response:

This plot suggests that the prediction model suffers from overfitting and early stopping as one method that can minimise overfitting is not effective in increasing the model's performance. To investigate the model's performance further, the error distribution from the model is created as it can be seen in Figure 8. From the error distribution, it is shown that most errors are created when the prediction is one class apart from the true labels. This leads to the idea of performance improvement by converting multiclass classification task into binary classification. In this case, rather than predicting a bridge's load capacity, the model only predicts whether a bridge's load capacity is higher or lower than a specific load level.

6. Figure 8 Is the error plot for training or test data?

Response:

Figure 8 describes the error distribution for testing data. We have included this information in section 4.4 (page 8) within our revised manuscript.

7. What are DL1, DL2 etc? These don't seem to be explained.

Response:

These are the datasets that are utilised on this study. LRx represents datasets that are used for load rating prediction models while DLx consists datasets for design load prediction models. All datasets are explained in section 3 and corresponding table has provided for explanation. The table below shows the datasets and the section where these datasets are described.

Dataset	Described in section	Corresponding Table
LR1, LR2, LR3, LR4, LR5, LR6 LR7, and LR8	3.4.1	Table 2
DL1, DL2, DL3, DL4, DL5, and DL6	3.4.1	Table 3
LR9, LR10, LR11, DL7, DL8, and DL9	3.4.2	Table 4
DL10, DL11, DL12, DL13, DL14, DL15, DL16, DL17, and DL18	3.4.3	Table 5

In the revised manuscript, the sections that describes these datasets along with the modification have been highlighted.

Reviewer: 2

Comments to the Author(s)

The method in this paper is the current practice of high-profile artificial intelligence. However, in bridge engineering, the application mainly focuses on bridge monitoring. It is not reliable to identify bridge load grade by deep learning. There are several questions for the author to answer:

Response:

Thank you for your insight. This paper's main outcome is not to provide a continuous monitoring but rather proposes an alternative way to obtain a rough estimate of the load capacity of a bridge, as an

experienced civil engineer would be able to do. As it has been mentioned in the introduction section of this manuscript, the proposed method is intended to be used in a condition when rapid assessment is necessary such as in post-disaster zones or when expertise is not available. An example is whether my aid delivery truck can potentially travel over this bridge or not, whereby only the lower bound of load capacity needs to be estimated. In these situations, performing detailed human assessment of the load rating activity might require time and resource that is not available.

1. Firstly, most of the identified images cannot reflect the full view of the bridge, how to ensure the accuracy of identification.

Response:

In section 4.2, we have reported the impact of image completion to the performance of the prediction model. As it is explained in section 3.4.3, this section compares the performance of prediction model trained using several datasets that are generated according to the image completion. These datasets are datasets consisting all images, datasets consisting images showing full view of bridges, and datasets consisting images showing incomplete view of bridges. In order to ensure the accuracy of the identification, the performance of the prediction model that are trained using these datasets are compared and reported in section 4.2.

2. Second, image recognition cannot judge the information inside the bridge, such as how to distinguish the reinforced concrete structure from the prestressed concrete structure. If not, how to ensure the accuracy only by image recognition.

Response:

Thank you very much for pointing this valuable feedback. To clarify the distinction between general load classification (this paper) and detailed health monitoring, we have revised the premise of our paper (introduction) to state more clearly: *"In this situation end users (i.e., aid workers and military) have to prioritise which bridges to inspect with limited resources. Therefore, we are motivated to classify broad bridge capacity values as a precursor to closer condition inspection."* We also go on to say that our future work intends to combine the proposed generalised classifier with conditional monitoring classification which we are currently working on.

At present, it is challenging to obtain some features that might be beneficial for capacity prediction visually. Therefore, it is challenging to distinguish these materials from an image while at the same time this can be one important feature that is required for this task. In order to tackle this issue, this important feature might be added as an additional input of the future prediction models. In this case, the prediction model will require images and these additional input (i.e. bridge material, bridge condition) in order to make a prediction. The revised manuscript has been modified to include the importance of bridge material (section 4.1.3 page 7 and conclusion page 9).

3. Thirdly, in the operation stage, the bearing capacity of the bridge structure will decline to different degrees, which is not considered in the identification results.

Response:

We have identified the structure condition (ie. Bearing capacity during operation) as one of the factors that produce uncertainty in our study. Similar to our previous response, the bearing capacity might not be detected visually, and it might influence the prediction accuracy, but as stated, our goal is to not initially find the bearing capacity. In our manuscript, the result from section 4.1.3 shows that the performance of design load prediction model that might not be affected by structure's condition outperforms the performance of load rating prediction model that might be influenced by structure condition. This might happen since this structure condition is not considered in the training process.

4. Fourth, the author uses the data from NBI database for deep learning. If the results of deep learning are used to identify a bridge in other countries, what is the recognition result.

Response:

Although we use bridges from America, the design of Bridge in all parts of the world will follow standard design code. If the prediction in our study is implemented in a country that that adopt US design code (AASHTO), then similar result can be obtained. However, if it is utilised in a country that uses other standard (ie. Eurocode) than it might introduce new source of uncertainty. The deep learning model can only learn from the data and it might only be as good as the data that we supply into the model.

5. Fifth, what are the feature points for bridge identification.

Response:

Some important features might include the bridge's span, bridge material, bridge's condition, and bridge type. However, we did not specify this feature in our study since we use deep learning and not feature based learning. Therefore, the features are learnt automatically through supervised learning process since one motivation of our study is reducing the need of expertise. The pre-trained network that we utilised in the study combines both feature extraction and classification function in one body. The important features are extracted automatically using the feature extractor part of the model and these features are employed by the classifier part to perform classification. This information has been added in the revised manuscript.

6. Sixth, whether the technology has been successfully implemented, if so, please provide.

Response:

For load capacity prediction purpose, as far as our knowledge, this is the first research that implement deep learning. To encourage other researchers in improving this research we have supplied all datasets and codes that we used on this research on Dryad server.

In general, deep learning has been widely implemented in other research domain and has become a state of the art for image processing purpose. Some successful applications include the image classification of a large number of classes [2-4], face recognition [5-6], damage detection in structures [7-10], and medical object classification [11-13]. In addition, big companies such as Google, Facebook, Microsoft, IBM, Yahoo!, and Adobe have now moved into deep learning for their image classification purpose [14].

We hope the revised manuscript is now suitable for publication. In addition, we hope we have made satisfactory responses to all reviewers. We look forward to hearing from you regarding our submission.

Sincerely,

Arya Pamuncak

References:

- [1] Kanan, C. and Cottrell, G.W., 2012. Color-to-grayscale: does the method matter in image recognition?. *PloS one*, 7(1), p.e29740.
- [2] Krizhevsky A, Sutskever I, Hinton G. ImageNet classification with deep convolutional neural networks. *Communications of the ACM*. 2017;60(6):84-90.
- [3] Simonyan K, Zisserman A. Very deep convolutional networks for large-scale image recognition. *arXiv preprint arXiv:1409.1556*. 2014 Sep 4.
- [4] Szegedy C, Liu W, Jia Y, Sermanet P, Reed S, Anguelov D, Erhan D, Vanhoucke V, Rabinovich A. Going deeper with convolutions. In *Proceedings of the IEEE conference on computer vision and pattern recognition 2015* (pp. 1-9).

- [5] Taigman Y, Yang M, Ranzato MA, Wolf L. Deepface: Closing the gap to human-level performance in face verification. In Proceedings of the IEEE conference on computer vision and pattern recognition 2014 (pp. 1701-1708).
- [6] Parkhi OM, Vedaldi A, Zisserman A. Deep face recognition. In BMVC 2015 Sep 7 (Vol. 1, No. 3, p. 6)
- [7] Cha Y, Choi W, Büyüköztürk O. Deep Learning-Based Crack Damage Detection Using Convolutional Neural Networks. *Computer-Aided Civil and Infrastructure Engineering*. 2017;32(5):361-378.
- [8] Protopapadakis E, Stentoumis C, Doulamis N, Doulamis A, Loupos K, Makantasis K et al. AUTONOMOUS ROBOTIC INSPECTION IN TUNNELS. *ISPRS Annals of Photogrammetry, Remote Sensing and Spatial Information Sciences*. 2016;III-5:167-174.
- [9] Zhang L, Yang F, Zhang YD, Zhu YJ. Road crack detection using deep convolutional neural network. In *Image Processing (ICIP), 2016 IEEE International Conference on* 2016 Sep 25 (pp. 3708-3712). IEEE.
- [10] Gopalakrishnan K, Khaitan S, Choudhary A, Agrawal A. Deep Convolutional Neural Networks with transfer learning for computer vision-based data-driven pavement distress detection. *Construction and Building Materials*. 2017;157:322-330.
- [11] Li Q, Cai W, Wang X, Zhou Y, Feng DD, Chen M. Medical image classification with convolutional neural network. In *Control Automation Robotics & Vision (ICARCV), 2014 13th International Conference on* 2014 Dec 10 (pp. 844-848). IEEE.
- [12] Hoo-Chang S, Roth HR, Gao M, Lu L, Xu Z, Nogues I, Yao J, Mollura D, Summers RM. Deep convolutional neural networks for computer-aided detection: CNN architectures, dataset characteristics and transfer learning. *IEEE transactions on medical imaging*. 2016 May;35(5):1285.
- [13] Tajbakhsh N, Shin J, Gurudu S, Hurst R, Kendall C, Gotway M et al. Convolutional Neural Networks for Medical Image Analysis: Full Training or Fine Tuning?. *IEEE Transactions on Medical Imaging*. 2016;35(5):1299-1312.
- [14] LeCun Y, Bengio Y, Hinton G. Deep learning. *Nature*. 2015;521(7553):436-444.

Appendix B

Dear Mr. Andrew Dunn,

We would like to thank you for providing us this opportunity to revise our manuscript titled Deep Learning for Bridge Load Capacity Estimation in Post-Disaster and -Conflict Zones to *Royal Society Open Science*. The feedback and suggestions offered by the reviewers have been constructive, and we appreciate the time and effort that you and the reviewers have dedicated for providing us this valuable feedback. We have addressed the reviewer comments in our revised manuscript, and we would like to provide response on the comments. Please find our responses below.

Reviewer: 1

Comments to the Author(s)

The authors have addressed most reviewer comments. The main concern remaining is the language. There are a number of grammatical errors and these can be fixed with a careful proofread or with assistance from a native English speaker.

Response:

Thank you very much for your constructive feedback. In the revised manuscript, we have conducted a thorough language check and made some improvements. In addition, the writing has been improved with the assistance from an experience writer who is a native speaker. We also have attached a signed letter from a native speaker of English.

A secondary concern is regarding the main focus of the paper. The emphasis appears to be on an approach to approximately but rapidly load rate a bridge in the aftermath of a major event. While this is mentioned to some degree, the authors could also make it clear that the proposed approach will not pick up localised or global damage but instead just attempt to quickly assess the load capacity based on a picture that provides information such as span length, width, etc. In fact, the information that needs to be in the picture can be discussed more in the study by relating to the performance of the approach and trying to come up with inferences on the information lacking in the pictures that were incorrectly interpreted.

Response:

We have added a new section in our revised manuscript called “**Analysis on the Model’s Uncertainties**” to include this information. This section contains an analysis of uncertainties as well as analysis on the important features required to enhance the performance of the prediction models.

Reviewer: 2

Comments to the Author(s)

I am glad to see your detailed answers to the questions raised. It is suggested that the author make a brief analysis of the causes of errors in a separate section.

Response:

Thank you for your insight. In our revised manuscript, we have added a new section “**Analysis on the Model’s Uncertainties**” to explain our analysis on the cause of errors on our proposed method.

We hope the revised manuscript is now suitable for publication. In addition, we hope we have made satisfactory responses to all reviewers. We look forward to hearing from you regarding our submission.

Sincerely,

Arya Pamuncak